# Gut Microbiota and Lipid Metabolism in Bullfrog Tadpoles: A Comparative Study Across Nutritional Stages

**DOI:** 10.3390/microorganisms13051132

**Published:** 2025-05-15

**Authors:** Zhilong Wang, Pengxiang Liu, Jun Xie, Huirong Yang, Guangjun Wang, Kai Zhang, Rui Shu, Zhifei Li, Jingjing Tian, Hongyan Li, Wenping Xie, Wangbao Gong, Yun Xia

**Affiliations:** 1Key Laboratory of Tropical and Subtropical Fishery Resource Application and Cultivation, Ministry of Agriculture, Pearl River Fisheries Research Institute, Chinese Academy of Fishery Sciences, Guangzhou 510380, China; 2School of Animal Science and Technology, Foshan University, Foshan 528225, China; 3College of Marine Sciences, South China Agricultural University, Guangzhou 510642, China; 4Guangdong Xingwa Agricultural Technology Co., Ltd., Zhaoqing 526070, China

**Keywords:** *Lithobates catesbeiana*, tadpole and nutritional stage, lipid metabolism, habitat, microbial community

## Abstract

Our study investigated the growth-related metabolic changes and microbial community dynamics during the early feeding stage of bullfrog (*Lithobates catesbeiana*) tadpoles. In this research, we examined the changes in fat accumulation patterns, as well as the levels of biochemical and enzymatic indicators and genes mRNA expression related to lipid metabolism across the endogenous, mixed, and exogenous nutritional stages of bullfrog tadpoles from a single mating pair. Simultaneously, we characterized the community structures of intestines, excreta, and water bodies during these stages using 16S rRNA high-throughput sequencing technology. Our findings reveal that fat accumulation in tadpole tissues gradually increases with the onset of feeding. Levels of alanine aminotransferase, aspartate aminotransferase, triacylglycerol, total cholesterol, non-esterified fatty acid, high-density lipoprotein cholesterol and low-density lipoprotein cholesterol show a significant increase in exogenous nutritional stages. The mRNA levels of lipid synthesis-related genes and lipid degradation-related genes increased gradually with the changes of nutritional stage. Significant differences were observed in microbial community characteristics among intestines, excreta, and water bodies across nutritional stages. Specifically, gut microbiota exhibited a lower similarity to water body microbiota but a higher similarity to excreta microbiota. Notably, the phyla Firmicutes and Actinobacteriota, and the genus *Cetobacterium* within the gut microbiota, increased with nutritional changes. A Spearman correlation analysis highlighted a significant correlation between gut microbiota composition and lipid metabolism markers, particularly a strong positive association between *Cetobacterium* and lipid-related parameters. These insights provide a theoretical foundation for nutritional interventions aimed at enhancing tadpole growth and survival rates.

## 1. Introduction

The bullfrog is renowned for its tender meat, rich nutrients, rapid growth, and robust environmental adaptability [1]. In recent years, bullfrog cultivation in China has seen rapid development, establishing itself as an emerging characteristic species in aquaculture, with the annual production reaching 700,000 tons in 2022 [2]. Upon hatching, amphibians initially depend solely on the nutrients within the yolk sac for their energy needs [3]. As the yolk sac is gradually depleted, they initiate external feeding to supplement their energy requirements. Eventually, once the yolk sac is entirely exhausted, their energy supply becomes fully dependent on the intake of external food sources [4]. The nutritional stages are delineated into endogenous, mixed, and exogenous three distinct stages according to the nature of the nutrients consumed [5]. Within the bullfrog production cycle, tadpole development represents a critical phase, with the successful transition to exogenous feeding being particularly vital for effective seedling cultivation [6]. Emerging evidence indicates that gut microbial communities mediate this transition through three key mechanisms: (1) the enhancement of nutrient assimilation efficiency, (2) the modulation of host metabolic pathways, and (3) the suppression of opportunistic pathogens—a processes well documented in fish and crustacean larvae [7,8,9].

Aquatic animal juveniles have complex interactions between their growth metabolism and microbial communities [10]. During the embryonic stage, microbes colonize the surface of fertilized eggs [11]. Following hatching, larvae initiate feeding, exposing their digestive systems to the environment and quickly becoming colonized by bacteria [12]. The colonization of gut microbiota benefits immune stimulation [13], enhances resistance to viral infections in larvae, and promotes nutrient metabolism, particularly carbohydrate and cholesterol metabolism [14]. Gut microbial communities significantly influence host development, growth, and health [15]. Prior studies have demonstrated that gut microbiota participate in energy homeostasis by regulating feeding, digestion, metabolism, and immune responses [16]. Lipids serve not only as energy stores but also as critical components of biological membranes, influencing various physiological functions and regulating host growth, development, and behavior [17]. In post-hatching larval stages of aquatic animals, lipid metabolism represents the predominant energetic pathway supporting growth and development, while simultaneously participating in organogenesis and maintaining physiological homeostasis, thereby exerting significant impacts on both developmental progression and survival rates [18]. Research on *Rana chensinensis* tadpoles has indicated that the gut microbiome plays a significant role in regulating lipid metabolism [17]. Once aquatic animals commence feeding, pathways such as bile acid biosynthesis, metabolism, and cholesterol catabolism become significantly enriched, likely influenced by gut microbiota [8].

The environmental microbial community within aquaculture systems plays a critical role in ecosystem material cycling and energy flow [19], which responds swiftly to environmental changes and serves as a vital biological indicator for monitoring shifts in water quality and disease outbreaks. Moreover, due to fish being in direct contact with water for extended stages, there exists a close interaction between intestinal microbiota and environmental microbiota. For instance, studies have observed a substantial overlap in operational taxonomic units (OTUs) between the intestine of tilapia and water microbiota, indicating effective microbial transmission from water to intestines [20]. Additionally, research on four cyprinid fish species revealed a strong correlation between fish intestinal microbiota and the surrounding water environment [21]. As an amphibian species, bullfrogs exhibit aquatic larval stages that share ecological niches with teleost fishes, particularly in lentic freshwater systems. The intimate link between intestinal microbiota and environmental microbiota allows them to be considered as an integrated whole. Changes in the environmental microbial community resulting from environmental stress can disrupt microbial network relationships within the system, potentially leading to the dysbiosis of host intestinal microbiota and disease outbreaks [22]. Hence, understanding the structure and diversity of microbial communities in aquaculture ecosystems, along with the interactions between environmental and host microbiota, is crucial for monitoring the health of cultured organisms, preventing and managing aquaculture diseases, and preserving ecosystem equilibrium [23].

In this context, this study aims to clarify the following: (1) How does the intestinal microbiota composition of bullfrogs shift in response to nutritional transition? (2) To what extent does environmental microbiota contribute to the assembly of gut microbial communities in bullfrogs? (3) What is the potential mechanistic link between gut microbiota structure and lipid metabolism regulation in developing bullfrogs? To investigate these interactions, microbial communities in the intestines, water, and excreta of bullfrogs at different nutritional stages within the aquaculture system, along with the distribution of lipids in body tissues and mRNA expression levels of lipid metabolic genes (with the endogenous stage serving as a control to isolate and quantify the independent contribution of initial feeding to lipid metabolism), were separately analyzed. We further investigated potential associations between gut microbial composition and host lipid metabolism through an integrated analysis of intestinal bacteria composition, lipid-related physiological and biochemical parameters, and lipid-related gene expression profiles. This research aims to provide deeper insights into the structure, characteristics, and functional roles of microbiomes in commercial aquatic species, enhancing our understanding of how they regulate host adaptability to environmental changes and to thereby improve aquaculture management practices.

## 2. Materials and Methods

### 2.1. Animal Ethics Statement

All animal experiments complied with the Code of Ethics of the Laboratory Animal Ethics Committee Pearl River Fisheries Research Institute, CAFS (ethical approval number: LAEC-PRFRI-2023-04-03, approved on 13 April 2023).

### 2.2. Aquaculture Management

The bullfrog embryos used in this study were obtained from a bullfrog hatchery in Guangzhou, China. To minimize the potential impact of host genotype variation, fertilized eggs were obtained from a single mating pair and placed in continuously dechlorinated and filtered tap water. Eggs were allowed to hatch naturally without any intervention. One day post-hatching, healthy bullfrog tadpoles (approximately 450 individuals) (Gosner Stage 23–24) were selected and randomly distributed into three rearing tanks (0.2 m × 0.2 m × 0.4 m) filled to three-quarters capacity with water transported from the original hatchery system, maintaining a stocking density of approximately 150 tadpoles per tank. After one day of acclimation, the feeding trial began. Throughout the rearing period, one-third of the water volume was replaced every 5 days using aerated, sterilized, and dechlorinated tap water. In the experiment, a small amount of commercial tadpole feed (Tongwei, Chengdu, China) was initially offered to observe the tadpoles’ feeding behavior. Once the tadpoles demonstrate feeding capabilities, they are fed twice a day at approximately 09:00 and 15:00, with meals consisting of about 4% of their body weight. The tadpoles in each aquarium are weighed every 5 days to adjust their feeding amount accordingly. The formal experiment lasted for 20 days. Throughout the experiment, water quality parameters were regularly monitored, including temperature (maintained at 27.0–29.0 °C), total ammonia nitrogen (<0.1 mg/L), dissolved oxygen (5.0–6.5 mg/L), and pH (7.0–8.0).

### 2.3. Sampling Procedure

In the experimental process, the method described by Wang et al. [5] was adopted, identifying and categorizing tadpoles into distinct nutritional stages based on their feeding ability and yolk sac regression. At the trial initiation (day 0), all tadpoles (Gosner Stage 23–24) were confirmed to be in the endogenous nutritional stage, as evidenced by a complete absence of oral feeding capability. Systematic behavioral monitoring demonstrated that by day 10, 100% of the tadpoles (Gosner Stage 25) had developed a feeding capacity while maintaining yolk sacs, satisfying diagnostic criteria for the mixed nutritional stage. Subsequent radiographic tracking (days 15–19) documented progressive yolk sac absorption, with complete resorption achieved by day 19 in all specimens (Gosner Stage 25), thereby establishing day 20 as the exogenous nutritional stage. Samples were collected on day 0 (0 d), day 10 (10 d), and day 20 (20 d) of the experiment based on the results of nutritional stage identification. During each sampling, tadpoles were anesthetized with MS-222 (60 mg/L; E10521, Sigma, St. Louis, MO, USA). Then, 2 tadpoles were randomly selected from each rearing tank (6 tadpoles in total), immediately frozen in liquid nitrogen, and stored at −80 °C for subsequent biochemical and enzymatic activity analyses. Similarly, another 6 additional tadpoles were selected and stored at −80 °C for subsequent RNA extraction. On days 0 and 10, 2 tadpoles per tank (6 in total) were fixed in 4% paraformaldehyde solution to prepare tissue sections for H&E staining. This process was repeated with another set of 6 tadpoles to prepare tissue sections for Oil Red O staining. However, for tadpoles at 20 d, due to their increased individual body size, it was technically unfeasible to prepare the intact-tadpole sections. Instead, 2 tadpoles per tank (6 in total) were dissected to obtain liver tissues, which were subsequently fixed in 4% paraformaldehyde solution for H&E staining. The same collection procedure was repeated, and liver tissues were prepared for Oil Red O staining.

For gut sample collection, 18 tadpoles were randomly selected from each tank, totaling 54 tadpoles. Following anesthesia with MS-222, the complete intestinal tract (including luminal contents) was aseptically dissected using sterile scissors and forceps. Intestinal samples from every 6 tadpoles were pooled together, resulting in a total of 9 replicate intestinal microbial samples per experimental group. The groups were labeled as G-E, G-M, and G-X, corresponding to the endogenous, mixed, and exogenous nutritional stages of the gut microbiota samples, respectively. Water samples were collected from each rearing tank 10 cm below the water surface, using 2 L water collection bags. From each bag, 200 mL of water was filtered through a 0.22 μm polycarbonate filter (47 mm in diameter; Whatman, Maidstone, UK), to eliminate non-target particles. The filters were stored at −80 °C, and 2 replicates of water samples were obtained from each tank, resulting in a total of 6 replicate samples for each group, labeled as W-E, W-M, and W-X (representing endogenous, mixed, and exogenous nutritional stages of water samples, respectively). Additionally, sterile 10 mL polyethylene pipettes were used to collect tadpole excrement from the bottom surface of each rearing tank. Following centrifugation (3000× *g* for 5 min at 4 °C), the supernatant was carefully removed, and the excrement samples were transferred into pre-labeled 2 mL sterile PE tubes, labeled as E-E, E-M, and E-X (representing endogenous, mixed, and exogenous nutritional stages of excrement samples, respectively), 3 samples were collected from each tank, resulting in a total of 9 replicate samples per group. All samples were stored at −80 °C for subsequent 16S rRNA high-throughput sequencing analysis.

### 2.4. Biochemical and Enzymatic Analysis

Accurately weigh the intact tadpole tissue samples (6 tadpole samples per group). For each gram of tissue (g), add 9 times the volume (mL) of physiological saline solution (1:9, *w*/*v*) and homogenize at 4 °C, and then centrifuge at 3000 rpm/min for 10 min at 4 °C. Take the supernatant and dilute it with physiological saline solution to prepare a 1% homogenate. Commercial kits (Jiancheng, Nanjing, China) were used to determine the triacylglycerol (TG), total cholesterol (TC), high-density lipoprotein cholesterol (HDL-C), low-density lipoprotein cholesterol (LDL-C), non-esterified fatty acid (NEFA), alanine aminotransferase (ALT), and aspartate aminotransferase (AST). The protein concentration was measured using the BCA method at 562 nm. This was repeated three times for each sample.

### 2.5. Histological Analysis

In our previous study, we detailed the methods for analysis using H&E staining and Oil Red O staining [24]. Tadpole tissues (3 groups, 6 tadpoles per group) were removed from 4% paraformaldehyde, dehydrated through various ethanol gradients, cleared with isopropanol, and subsequently embedded in paraffin. Sections of 5 μm thickness were cut using a rotary microtome (RM2235, Leica, Wetzlar, Germany) and mounted on slides. Following standard histological protocols, paraffin sections of each tadpole tissue sample were stained with H&E stains. Histological samples were observed and photographed using a vertical microscope from Leica biosystems in Wetzlar, Germany. The paraffin sections were made by Hubei BIOS Biotechnology Co., Ltd. (Wuhan, China).

Tadpole tissues fixed in paraformaldehyde were stained with Oil Red O. Firstly, the slices were washed with PBS three times at 5–10 min each. Then, the slices were washed in 60% isopropyl alcohol for 5 min and stained with Oil Red O dye solution (prepared with isopropyl alcohol) for 15 min away from light. The slices were then stained with hematoxylin dye solution for 5 min. Finally, the tablets were differentiated and washed with 1% hydrochloric acid, alcohol, and water in succession, in the same order, and sealed with glycerin. They were then observed using a microscope (Leica, Wetzlar, Germany).

### 2.6. qRT-PCR Analysis

The total RNA of the intact tadpole tissues (3 groups, 6 tadpoles per group) were isolated using the Trizol reagent (Invitrogen, Waltham, MA, USA). RNA integrity was determined in 1% agarose gel. RNA concentration and purity were then determined using an Implen nanophotometer (Implen Inc., Westlake Village, CA, USA) with A260 and A280 (A260:A280 ≥ 1.8 and ≤2.0). cDNA was obtained using the cDNA synthesis kit (Takara, San Jose, CA, USA). Appendix A presents the primer sequences that were used in this experiment. Prior to qRT-PCR experiments, primer specificity and efficiency were validated. Standard curves and correlation coefficients were determined by constructing standard curves using serial dilutions of cDNA. All standard curves exhibited correlation coefficients greater than 0.99, and PCR efficiencies for each primer were between 1.97 and 2.02. β-actin was applied as the reference genes. Quantitative real-time PCR was carried out in triplicate on a Light Cycler^®^96-Time PCR instrument (Roche, Rotkreuz, Switzerland). The 20 μL reaction systems comprised of diluted first-strand cDNA product (2 μL), 2 × SYBR Premix Ex TaqII (10 μL), forward/reverse primer (0.8 μL of each), and sterilized ddH_2_O (6.4 μL). The qRT-PCR cycling procedure were set as follows: 95 °C for 5 min, 45 cycles of 95 °C for 15 s, and 60 °C for 1 min. Negative controls were established by using the non-cDNA and DNase-treated non-reverse transcribed tissue RNA samples. Pfaffl’s mathematical model was used to calculate the relative gene expression [25].

### 2.7. Microbiological Community Analysis

Using the EZNA^®^ Soil DNA Kit (Omega, Irving, TX, USA) following the manufacturer’s instructions, microbial genomic DNA was extracted. The 16S rRNA gene V3–V4 region was amplified using the forward primer 338F (5′-ACTCCTACGGGAGGCAGCA-3′) and reverse primer 806R (5′-GGACTACHVGGGTWTCTAAT-3′). The amplification program was performed in a PCR system (GeneAmp 9700, ABI, Foster City, CA, USA) and further detail can be found in a previous study [26]. Sequencing was performed on an Illumina MiSeq platform (Illumina, San Diego, CA, USA), and this part of the work was performed by Majorbio Technology. All the sequence raw datasets have been deposited in the NCBI Sequence Read Archive with the Bio-Project accession number PRJNA1133824.

According to QIIME (Version 1.9.1), the raw tags underwent quality filtering, and optimized sequences were clustered into OTUs at a 97% similarity. The abundance of each OTU was determined. Sparse curves, alpha diversity metrics (including Simpson, Shannon, Chao1, and Pielou_e indices), hierarchical clustering and principal coordinate analysis (PCoA) based on Bray–Curtis distance, linear discriminant analysis effect size (LEfSe), neutral community model (NCM), and null model (999 randomizations) were computed using R software (version 3.3.1) on the Majorbio Cloud platform (https://cloud.majorbio.com/) [27]. In reference to the method by Meng et al. [28], a Spearman correlation analysis was employed to investigate correlations between the microbiome and host metabolic indicators, with significance set at *p* < 0.05.

### 2.8. Statistical Analysis

Data analysis was performed using SPSS 22.0 software (SPSS Inc., Chicago, IL, USA). All data were expressed as the mean ± standard error of the mean (SE). One-way ANOVA analysis was performed, followed by Duncan’s multiple comparisons between groups. A value of *p* < 0.05 indicates a statistically significant difference.

## 3. Results

### 3.1. Histological Observations of Organisms at Different Nutritional Stages

In the Oil Red O sections of the intact tadpoles from the 0-d and 10-d groups, no significant lipid droplets were observed, and no obvious lipid droplets were seen in the fibrous structures of the tadpole livers; whereas, in the tadpole livers of the 20-d group, relatively prominent lipid droplets (red) and nuclei (blue) were evenly distributed throughout the tissue (Figure 1).

### 3.2. Differences in the Biochemical and Enzymatic Indicators of Bullfrog Tadpoles

As shown in Table 1, in the 20-day group, all biochemical and enzymatic indicators were significantly higher than those in the 0-d and 10-d groups (*p* < 0.05). The TG and NEFA contents of the 10-d group were significantly higher than those of the 0-d group (*p* < 0.05), while there were no significant differences in ALT, AST, TC, HDL-C, and LDL-C between the 0-d and 10-d groups (*p* > 0.05).

### 3.3. Differences in Lipid Metabolism-Related Gene Expression in Bullfrog Tadpoles

The relative expression levels of lipid synthesis-related genes (*ppar-γ*, *fas*, *dgat1*) and lipid breakdown-related genes (*ppar-α*, *hsl*, *cpt1*, *acox1*) in bullfrog tadpoles are shown in Figure 2. The mRNA levels of each lipid metabolism-related gene showed an upward trend. Among them, compared with the 0-d group, the mRNA levels of lipid synthesis genes in the 20-d group all increased (*p* < 0.05), and the mRNA levels of lipid breakdown genes except *acox1* were also significantly upregulated (*p* < 0.05). There were no significant differences in the mRNA expression levels of each lipid metabolism-related gene between the 0-d and 10-d groups (*p* > 0.05). Compared with the 10-d group, the mRNA expression levels of lipid synthesis genes *fas* and *dgat1* in the 20-d group were significantly increased (*p* < 0.05), while the mRNA expression level of *ppar-α* was not significant between the two groups (*p* > 0.05). The mRNA expression levels of lipid breakdown genes *hsl* and *ppar-γ* in the 20-d group were significantly upregulated (*p* < 0.05), while there were no significant differences in the mRNA expression levels of *acox1* and *cpt1* between the two groups (*p* > 0.05).

### 3.4. Sequencing Data and Diversity Analysis

From a total of 63 samples (23 gut samples, 24 excrement samples, and 16 water samples), 3,939,919 optimized sequences were obtained, resulting in 1,643,124,600 high-quality bases after filtering. These sequences were clustered into 5239 OTUs at a 97% sequence similarity. The rarefaction curve based on OTU-level Shannon indices showed that the sequencing coverage for each sample nearly reached saturation, indicating a good overall sequencing coverage (Appendix A). There were 167 OTUs shared across all samples, with OTU richness showing the trend of excrement > gut > water (Figure 3B). In both water and excrement microbiomes, the number of OTUs decreased gradually with feeding stages, whereas in intestinal microbiomes, OTU numbers were lowest during the mixed nutritional stage and highest during the exogenous nutritional stage.

To comprehensively assess the α-diversity of the microbial communities, five diversity indices were calculated in this study. Chao1 was used to estimate species richness, Shannon and Simpson indices were used to estimate species diversity, the Pielou_e index was used to estimate community evenness, and the Goods_coverage index was used to estimate species coverage (Figure 3A). Across all samples, Goods_coverage was consistently no less than 99.0%, indicating that the sequencing depth adequately covered most microbial species. Shannon, Chao1, and Pielou_e indices of water microbiomes showed significant differences among different feeding stages (*p* < 0.05), with these indices decreasing gradually with feeding stages. Shannon and Chao1 indices of intestinal microbiomes during the mixed nutritional stage were significantly lower than during the endogenous nutritional stage (*p* < 0.05), but increased significantly during the transition to the exogenous nutritional stage (*p* > 0.05); the Pielou_e index showed no significant differences among different feeding stages (*p* > 0.05). The Chao1 index of the excrement showed a significant decreasing trend with feeding stages (*p* < 0.05), and Shannon and Pielou_e indices showed the highest values during the mixed nutritional stage and the lowest values during the exogenous nutritional stage.

### 3.5. An Analysis of Microbial Community Composition

The dominant microbial phyla in the gut and excrement of bullfrogs are similar, with those having a relative abundance exceeding 25.00% comprising Proteobacteria, Bacteroidota, Firmicutes, and Fusobacteriota (Figure 4A). Notably, the prevalence of Fusobacteriota in both the gut and excrement samples demonstrates a consistent increase, attaining respective abundances of 29.69% and 20.00% during the exogenous nutrition phase, thereby emerging as the third and second most dominant phyla, respectively. The dominant phyla in water microbial communities are Proteobacteria, Bacteroidota, and Actinobacteriota, distinctly different from those in the gut and excrement. During the exogenous nutritional stage, the abundance of Actinobacteriota increases to 53.19%, replacing Proteobacteria as the primary dominant phylum.

At the genus level, dominant genera in the excrement of the endogenous and mixed nutritional stage include *Bacillus* (34.81% and 8.64%) and *Rhodobacter* (6.72% and 9.32%) (Figure 4B). In contrast, during the exogenous nutritional stage, the dominant genus is primarily *Cetobacterium* (29.69%), with other genera accounting for a less than 10% abundance. The composition and abundance of the dominant genera in the gut vary significantly across different nutritional stages. During the endogenous nutritional stage, dominant genera include *Sphingobacterium* (15.51%) and *Bacillus* (4.18%), with *Sphingobacterium* sharply declining to less than 0.1% during the mixed and exogenous nutritional stages. The dominant genera during the mixed nutritional stage transition to *Bacteroides* (16.54%) and *Cetobacterium* (10.79%), with *Cetobacterium* gradually increasing in abundance throughout the nutritional transition, becoming the primary dominant genus (17.00%) during the exogenous nutritional stage.

The dominant genera in water show distinct differences from those in excrement and the gut. *Sediminibacterium* (18.00%) dominates during the endogenous nutritional stage; while during the mixed and exogenous nutritional stages, *Aurantimicrobium* (24.23% and 50.57%), *Acinetobacter* (16.54% and 4.30%), and *Polynucleobacter* (10.61% and 12.56%) increase in abundance, becoming new dominant genera.

Furthermore, the Spearman correlation analysis was conducted to explore potential correlations between the dominant phyla and genera (top 10 in relative abundance) of gut microbiota and organism metabolic indicators. The results indicate significant correlations (Figure 4C,D). Among them, *Cetobacterium* showed a significant positive correlation with the expression levels of TG, TC, HDL-C, LDL-C, and lipid metabolism genes (*p* < 0.05).

### 3.6. An Analysis of the Similarities and Differences of Microbial Communities

The hierarchical clustering and PCoA analysis revealed consistent findings, with gut and fecal samples clustering closely together, while water samples formed distinct clusters separate from both excrement and gut samples (Figure 5A,B). Within water samples, clustering was less cohesive, as supported by Adonis analysis (Appendix A), indicating significant (*p* < 0.05) differences in the microbial community structure across the three nutritional stages. Specifically, during the endogenous nutritional stage, there was notable segregation between excrement and gut samples compared to other nutritional stages. In contrast, during mixed and exogenous nutritional stages, excrement and gut samples clustered closely together. Furthermore, excrement and gut samples from different nutritional stages both showed a significant separation, tending to cluster separately. Combined with Adonis analysis, this indicates significant differences in the community structure among excrement samples from three different nutritional stages, as well as among gut samples.

The linear discriminant analysis (LDA) effect size (LEfSe) was utilized to compare indicator taxa at the genus level across different habitats during various nutritional stages (Figure 5C). The histogram of LDA scores illustrates significant abundance differences among habitats across the three nutritional stages. For the gut, LDA scores indicate enrichment of eight genera such as *Sphingobacterium* and *Comamonas* in G-E (*p* < 0.05); nine genera including *Bacteroides*, *Parabacteroides*, and *Akkermansia* in G-M (*p* < 0.05); and eight genera like *Aeromonas* and *Cetobacterium* in G-X (*p* < 0.05). In water samples, eight genera such as *Sediminibacterium* are enriched in W-E (*p* < 0.05); *Acinetobacter*, *Flavobacterium*, *Legionella*, and *Exiguobacterium* in W-M (*p* < 0.05); and *Aurantimicrobium*, *Polynucleobacter*, and *Reyranella* in W-X (*p* < 0.05). Regarding excrement samples, *Bacillus*, *Gemmobacter*, *Dinghuibacter*, and *Romboutsia* are enriched in S-E (*p* < 0.05); five genera including *Rhodobacter* in S-M (*p* < 0.05); and four genera such as *Cetobacterium* in S-X (*p* < 0.05).

### 3.7. Assembly Processes of Microbial Communities in Different Habitats

Using the OTU dataset, the assembly mechanisms of microbial communities in different habitats were studied using NCM and null models (Figure 6A). NCM explained all of the results based on the OTU dataset; each habitat sample and all samples had a good fit (all R^2^ values ranged from 0.395 to 0.566), indicating that stochastic processes were very important for the formation of microbial community assembly under different habitats. Moreover, the Nm and m values of excreta were higher than those of intestines and water bodies, indicating that bacterial species dispersal in excrement was higher than in other habitats. In this experiment, the relative roles of deterministic processes (homogeneous and heterogeneous selection) and stochastic processes (dispersal limitation, homogeneous dispersal, and drift) in microbial communities of different habitats were quantified. Most beta nearest taxon index (βNTI) values in water bodies were >−2 and <2, indicating that bacterial community assembly in water bodies was primarily formed by stochastic processes (Figure 6B). Against the backdrop of stochastic processes, we determined that the main steps in the formation of bacterial communities in water bodies were dispersal limitation and drift, accounting for 35.93% and 34.37%, respectively (Figure 6C). In the intestines, both stochastic and deterministic processes were dominant; within the deterministic processes, heterogeneous selection was the most important process controlling intestinal (49.53%) microbial communities. Compared to the intestines, microbial communities in excreta were controlled by a smaller proportion of deterministic processes, with stochastic processes being dominant, where dispersal limitation (37.15%) was the main advantageous process, followed by heterogeneous selection (37.15%) in deterministic processes, which was a non-dominant process, and then 20.48% drift.

## 4. Discussion

Observations of Oil Red O-stained liver sections in this experiment showed that no significant lipid droplets were observed in the tadpole liver tissues at 0 days and 10 days. Importantly, the 0-day data served as a critical endogenous baseline control, allowing us to disentangle the effects of nutritional stage transitions and the subsequent feeding activity on lipid metabolism. However, at 20 days, distinct and evenly distributed lipid droplets were evident throughout the tissue. This indicates that bullfrog tadpoles improve their tissue fat accumulation through feeding activities. TG, TC, and NEFA are predominantly synthesized and stored in the liver and represent the main components of hepatic fat. These levels can reflect the organism’s lipid metabolism status [29]. HDL-C, a beneficial form of cholesterol, transports cholesterol from peripheral tissues back to the liver [30], whereas LDL-C transports cholesterol from the liver to peripheral tissues, exhibiting functions opposite to HDL-C [31]. In this study, the levels of TG, TC, NEFA, HDL-C, and LDL-C in tadpole organisms gradually accumulated, indicating that tadpoles improve their lipid metabolism levels through changes in their feeding activity and nutritional stages. AST and ALT are two highly active aminotransferases in the liver, playing crucial roles in animal protein metabolism. To some extent, they can reflect the organism’s protein metabolism level [32]. In this research, the levels of ALT and AST in the organism gradually increased, suggesting that tadpole larvae’s amino acid metabolism can be improved through their feeding activities.

Transcription factors and enzyme gene expression levels related to lipid metabolism regulation are crucial hubs influencing lipid metabolism [33]. Lipid metabolism includes synthesis and breakdown: the former converts acetyl-CoA into fats for storage through a series of reactions, while the latter breaks down fats into glycerol and fatty acids for β-oxidation energy supply [29]. Key transcription factors and enzyme genes include lipid synthesis-related genes (*ppar-γ*, *fas*, and *dgat1*) and lipid breakdown-related genes (*ppar-α*, *hsl*, *cpt1*, and *acox1*). *Ppar-γ* is a transcription factor that regulates fatty acid synthesis by promoting *fas* gene expression [34]. *Fas*, as the rate-limiting enzyme in fatty acid synthesis, promotes fatty acid synthesis [35]. *Dgat1*, a crucial acyltransferase gene in triacylglycerol synthesis, catalyzes the final step of TG synthesis [36]. This study found that with the progression of feeding stages and nutritional transitions, the expression of lipid synthesis-related genes (*ppar-γ*, *fas*, and *dgat1*) in bullfrog tadpoles significantly increased, indicating that feeding activities markedly improve larval lipid synthesis metabolism. *Ppar-α* can promote fatty acid oxidation by activating *hsl* and *cpt1*, thereby improving lipid metabolism and reducing the total cholesterol and triacylglycerol levels [37]. *Acox1*, a downstream gene regulated by *ppar-α*, is a core enzyme involved in fatty acid β-oxidation and initiates the entire process, playing a crucial role in fatty acid breakdown [38]. *hsl* and *cpt1* are critical enzymes in fatty acid β-oxidation; *hsl* breaks down triacylglycerols into fatty acids for cellular utilization [39], while *cpt1* facilitates the transport of fatty acyl-CoA, a metabolite of fatty acid metabolism, into mitochondria for β-oxidation [40]. Our study found that with the progression of feeding activities and nutritional transitions, the expression levels of genes related to lipid synthesis and breakdown in the organism were significantly upregulated, suggesting that bullfrog tadpoles can improve their lipid metabolism through external feeding.

The study of microbial community characteristics is crucial for improving aquatic environments for aquatic organisms. Aquatic animal juveniles can adapt changes in developmental stages and nutritional types by controlling the ecological succession of intestinal flora [41]. The changes in microbial communities in aquatic animal juveniles are complex and generally considered to depend on microorganisms from eggs, water, and live feeds [42]. During the endogenous nutritional stage, the nutrients required for development come entirely from the yolk sac. At this time, the microbial community may consist of specific symbiotic microbes inherited from the embryonic stage of bullfrogs. The microbiota during the mixed and exogenous nutritional stages are closely related to feeding activities, which may lead to succession in the intestinal microbiota of bullfrog larvae. Results from Shannon and Chao1 indices of intestinal microbiota show that after tadpoles transition to a mixed nutritional stage, microbial diversity and richness decrease, while they increase after transitioning to an endogenous nutritional stage. We speculate that the intestinal microbiota of bullfrogs undergo adaptive changes to accommodate shifts in nutritional types, resulting in a notable decrease followed by an increase in diversity and richness. Additionally, the growth and development of bullfrog larvae themselves are important factors influencing changes in intestinal microbiota. Research on zebrafish has found continuous changes in intestinal microbial communities during fish growth and development, which are not necessarily linked to the water environment and feeding [43]. Hierarchical clustering and PCoA analysis results reveal that bacterial communities inhabiting the same habitat have closer phylogenetic relationships, while communities from different habitats show significant structural differences. Studies indicate that bacterial taxa with close genetic distances have more similar ecological niches [44]. Therefore, under different environmental pressures, different habitats often harbor bacterial communities with distinct ecological niches [45]. This mechanism may partially account for the observed divergence in microbial communities among the gut, water, and excrement in our study.

Community structure is a key factor influencing microbial functions. This study observed significant differences in gut microbiota composition during different nutritional stages, particularly at the phylum level, with notable variations in the abundance of Firmicutes and Bacteroidetes. Firmicutes are known for their capabilities in carbohydrate metabolism, facilitating the host’s utilization of feed resources [46]. Previous research suggests that the coexistence of Firmicutes and Bacteroidetes can jointly promote energy absorption or storage in the host, with a higher Firmicutes/Bacteroidetes ratio potentially enhancing these processes [46,47]. In this study, the Firmicutes/Bacteroidetes ratio in the host gut microbiota was highest during the exogenous nutritional stage, indicating an increased potential for energy absorption or storage, which aligns with the results of lipid metabolism markers and related gene expression levels. At the genus level, significant differences were observed in the abundance of *Cetobacterium*, *Bacteroides*, and *Bacillus*. *Cetobacterium*, known for its ability to colonize fish intestines [48], may contribute to improved amino acid transport and metabolism rates [49]. We found that the abundance of *Cetobacterium* increased gradually with the changes in nutritional stages. The Spearman correlation analysis also indicated a significant positive correlation (*p* < 0.05) between *Cetobacterium* abundance and levels of organismal TG, TC, HDL-C, and LDL-C, as well as expression levels of lipid synthesis-related genes, underscoring its significant role in tadpole growth, development, and lipid metabolism. *Bacillus*, a common probiotic in aquatic animal intestines [50], exhibited an increasing abundance with feeding activities and changes in nutritional stages, reflecting the host’s adaptation to growth, development, and the stability of intestinal microbiota during feeding. Additionally, *Bacteroides* bacteria can utilize polysaccharides and release anti-inflammatory substances that enhance the immune system’s capabilities [51], its abundance also showed a gradual increase with feeding activities and changes in nutritional stages.

Understanding the interactions between gut microbiota and environmental microbiota in aquaculture systems is crucial for developing effective microbial management strategies to promote the health of cultured organisms. Our findings confirm significant differences in the composition of microbial communities among water, excrement, and gut microbiota. Previous studies have also demonstrated substantial disparities in microbial community structures between aquaculture system water, sediments, and intestinal microbiota [52], consistent with our research results. Actinobacteriota and Fusobacteriota are microbial phyla whose abundance gradually increases in water and excreta, respectively. Actinobacteria can degrade carbohydrates and proteins, inhibit pathogen activity, and contribute to intestinal homeostasis regulation [53]. Fusobacteriota, predominantly represented by the genus *Cetobacterium* in excrement, exhibit similar distribution trends to intestinal microbiota, suggesting that Fusobacteriota and *Cetobacterium* in tadpole excreta may originate primarily from the intestinal microbial community. LEfSe analysis confirms *Cetobacterium* as the most significantly abundant indicator genus related to gut and excrement in exogenous nutritional stage. Known as a potential probiotic [54], *Cetobacterium* highlights the intestinal microbiota’s potential to interact with the surrounding environment and selectively enrich specific taxonomic groups, crucial for host health.

The assembly of microbial communities is crucial for understanding the mechanisms that regulate ecosystem-level functions. Compared to water, the NCM exhibited a better fit for gut and excrement microbial communities, suggesting that stochastic processes such as ecological drift and passive diffusion play a more significant role in shaping these communities. This observation aligns with research indicating that stochastic processes have a larger impact on vertebrate gut microbiota assembly [55], supporting our findings. Using null models, we quantified the relative importance of different ecological processes affecting microbial community assembly. Our analysis revealed that heterogeneous selection is the primary ecological process influencing gut microbiota assembly, followed by drift. In contrast, stochastic processes predominantly govern microbial community assembly in water and excreta. Previous studies have also shown that after the initial colonization in the gut, microbial proliferation and abundance are influenced by both deterministic (selection) and stochastic (drift) processes [56], which corroborates our findings. Moreover, microbial adaptation to different habitats encounters diverse pressures, underscoring the significant role of habitat filtering in community establishment, which favors species with similar functional traits in a given environment [57]. Conversely, heterogeneous selection and stochastic processes create broad ecological niches that allow species from different lineages or with distinct functional traits to coexist [58]. In this study, microbial community assembly processes across different habitats were predominantly characterized by heterogeneous selection and stochastic processes, which likely contribute to the regulation of microbial community stability.

An important consideration in interpreting the results of this study is the use of bullfrog tadpoles originating from a single mating pair. This design choice aimed to reduce genetic heterogeneity among subjects, thus controlling for potential background genetic effects on the observed outcomes. However, we recognize this limits the extent to which our findings can be generalized, as the genetic diversity of the wider species population is not captured. Therefore, while informative for this specific lineage, extrapolation to the entire species requires caution. Future investigations employing subjects from multiple, genetically diverse pairings are planned to validate these findings and ascertain the prevalence of these traits across the broader bullfrog population.

## 5. Conclusions

This study examines the lipid metabolism characteristics and gut microbiota dynamics of bullfrog (*Lithobates catesbeianus*) tadpoles across endogenous, mixed, and exogenous nutritional stages, highlighting the significant impact of nutritional transitions on host metabolism and microbial communities. The results indicate that as nutritional stages shift, lipid accumulation in tadpole tissues increases, accompanied by a significant upregulation of lipid metabolism-related genes (e.g., *ppar-γ*, *fas*, and *hsl*). Concurrently, the relative abundance of Firmicutes, Actinobacteriota, and *Cetobacterium* in the gut microbiota rises markedly. A Spearman correlation analysis reveals a significant positive correlation between *Cetobacterium* and lipid metabolism indicators (e.g., TG, TC, and HDL-C), suggesting that gut microbiota may indirectly regulate host lipid metabolism through the metabolic activities of specific microbial taxa. Moreover, significant differences are observed in the microbial composition between the surrounding water, feces, and gut microbiota. The assembly of these microbial communities is primarily governed by heterogeneous selection and stochastic processes, offering theoretical insights into the stability of microbiota in aquaculture systems. Our findings offer new insights into how microbial communities in aquaculture systems respond to changes in tadpole metabolic characteristics. However, the specific mechanisms underlying these associations warrant further exploration, which we aim to investigate using metagenomics and metabolomics technologies.

## Figures and Tables

**Figure 1 microorganisms-13-01132-f001:**
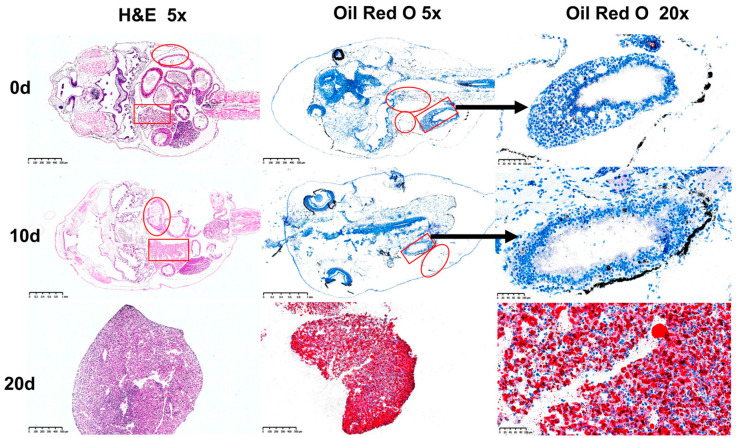
Results of the hematoxylin–eosin staining and Oil Red O staining of tadpoles. The tissue indicated by the red rectangle represents the liver, and the area marked by the red ellipse indicates the intestine.

**Figure 2 microorganisms-13-01132-f002:**
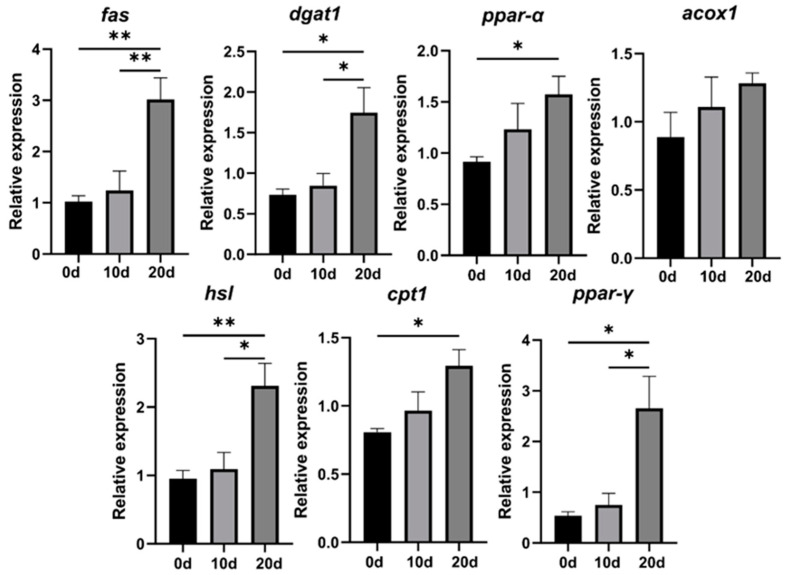
Differential mRNA expression levels of lipid metabolism-related genes in bullfrog tadpoles. Data are expressed as the mean ± SE (*n* = 6). Asterisks denote statistical differences between groups, and * 0.01 < *p* < 0.05, ** 0.001 < *p* ≤ 0.01. Note: peroxisome proliferators activated receptor-γ (*ppar-γ*); peroxisome proliferators activated receptor-α (*ppar-α*); fatty acid synthase (*fas*); diacylglycerol O-acyltransferase 1 (*dgat1*); hormone-sensitive lipase (*hsl*); carnitine O-palmitoyltransferase-1 (*cpt1*); acyl-CoA oxidase (*acox1*).

**Figure 3 microorganisms-13-01132-f003:**
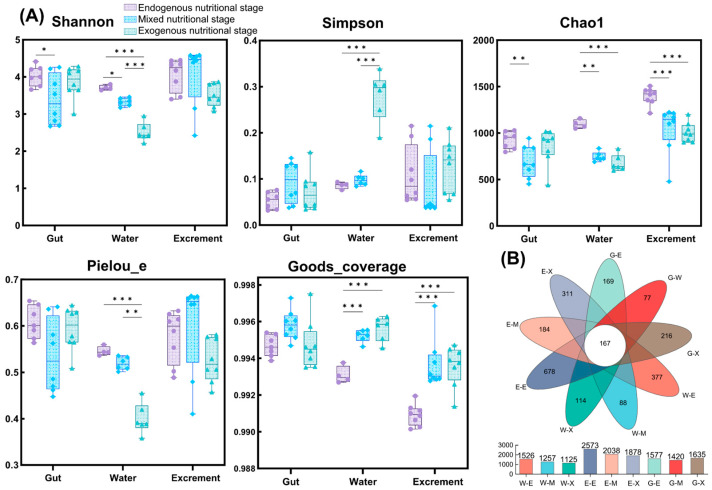
A comparative analysis of bacterial community α-diversity across different habitats at the OTU level. (**A**): Shannon, Simpson, Chao1, Pielou_e, and Goods_coverage are shown. The analysis results are displayed with boxplots. Boxes cover the interquartile range (IQR) and the line inside the box denotes the median. Whiskers represent the lowest and highest values within 1.5× IQR. (**B**): A Venn diagram showing the overlap of OTUs between different groups. Asterisks denote statistical differences between groups, and * 0.01 < *p* < 0.05, ** 0.001 < *p* ≤ 0.01, *** *p* ≤ 0.001.

**Figure 4 microorganisms-13-01132-f004:**
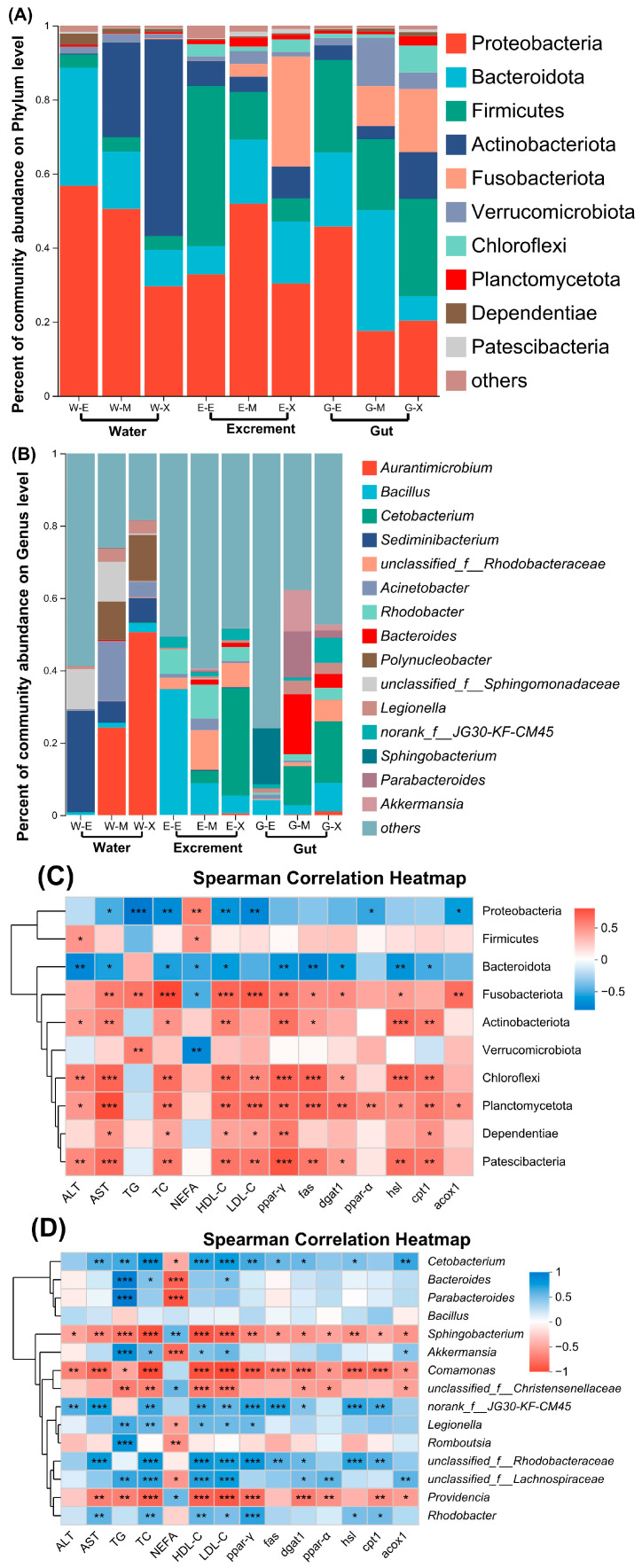
Microbial community composition. (**A**): Microbial community composition at the phylum level. (**B**): Microbial community composition at the genus level. (**C**): A correlation analysis between the top 10 most abundant bacterial phyla in the gut microbiota and metabolic markers. (**D**): A correlation analysis between the top 10 most abundant bacterial genera in the gut microbiota and metabolic markers. Asterisks denote statistical differences between groups, and * 0.01 < *p* < 0.05, ** 0.001 < *p* ≤ 0.01, *** *p* ≤ 0.001.

**Figure 5 microorganisms-13-01132-f005:**
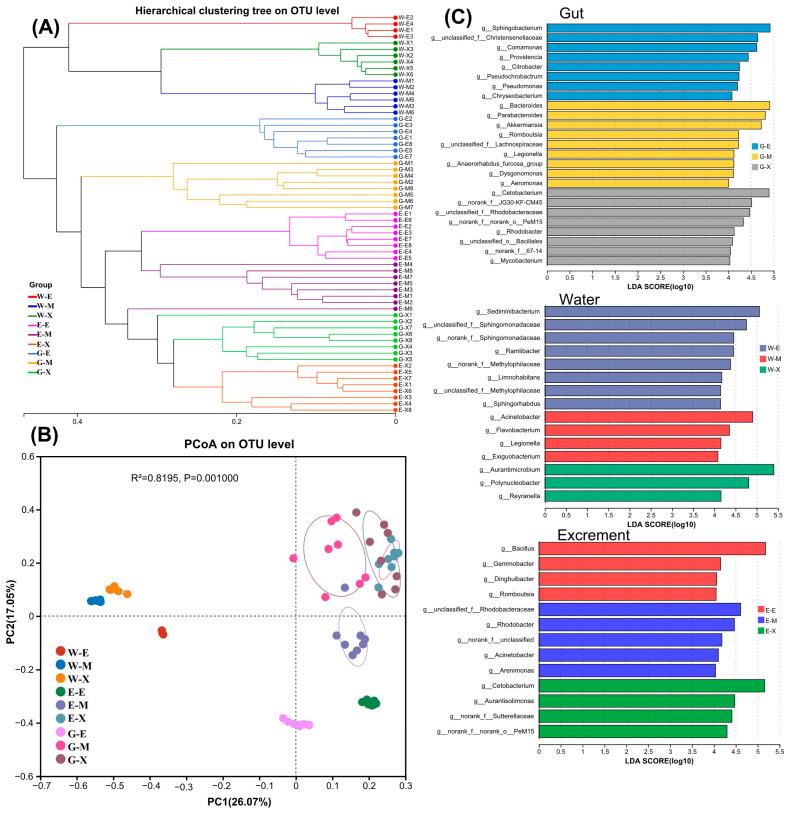
An analysis of the similarities and differences of bacterial communities in different habitats. (**A**): A hierarchical clustering of all the microbiological community analysis samples according to the Bray–Curtis distance metric. (**B**): A PCoA analysis based on the Bray–Curtis distance. (**C**): A linear discriminant analysis (LDA) effect size (LEfSe) analysis in the gut, excrement, and water (LDA scores > 4 and *p* <  0.05 are shown).

**Figure 6 microorganisms-13-01132-f006:**
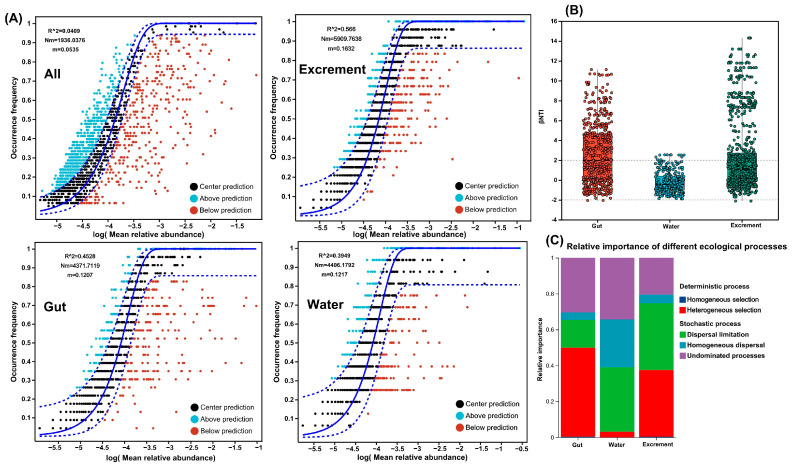
Estimating the stochasticity of the community assembly of different habitats. (**A**): The fit of the neutral community model (NCM) of community assembly. Solid blue lines represent the best fit of the neutral community model, dashed blue lines represent the 95% confidence interval around the model prediction, OTUs that occur within the predicted range are shown in black, and OTUs that occur more (blue) or less (red) than predicted by the NCM shown in different colors. R^2^ represents the fit degree of the model, and Nm represents the product of community size and migration times, and m represents the community-level migration rate, which is uniform for each community member (regardless of species). (**B**): A null model was used to analyze the assembly process of main microbial communities in different habitats. (**C**): Null model showing the contributions of different ecological processes to the assembly of the microbial community.

**Table 1 microorganisms-13-01132-t001:** Biochemical and enzymatic indicators of bullfrog tadpoles.

Items	0 d	10 d	20 d
ALT (U/gprot)	29.90 ± 8.35 ^a^	28.02 ± 8.53 ^a^	55.41 ± 10.97 ^b^
AST (U/gprot)	15.62 ± 6.41 ^a^	18.70 ± 7.95 ^a^	35.01 ± 7.67 ^b^
TG (mmol/gprot)	0.08 ± 0.01 ^a^	0.31 ± 0.05 ^b^	0.44 ± 0.15 ^c^
TC (mmol/gprot)	0.07 ± 0.01 ^a^	0.18 ± 0.01 ^a^	0.38 ± 0.05 ^b^
NEFA (mmol/gprot)	0.09 ± 0.01 ^a^	0.21 ± 0.02 ^b^	0.27 ± 0.01 ^c^
HDL-C (mmol/gprot)	0.02 ± 0.01 ^a^	0.07 ± 0.02 ^a^	0.27 ± 0.08 ^b^
LDL-C (mmol/gprot)	0.01 ± 0.00 ^a^	0.05 ± 0.01 ^a^	0.39 ± 0.05 ^b^

Note: Data are expressed as the mean ± SE of six replicates (*n* = 6). Values of the same row with different letters were significantly different (*p* < 0.05).

## Data Availability

The original contributions presented in this study are included in the article/Appendix A. Further inquiries can be directed to the corresponding authors.

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
