# Peer review of "Gut Microbiota and Lipid Metabolism in Bullfrog Tadpoles: A Comparative Study Across Nutritional Stages"

_microorganisms, 2025, doi:10.3390/microorganisms13051132_

Round 1
Reviewer 1 Report
Comments and Suggestions for Authors
In this manuscript, the authors track the changes in microbiome, and lipid metabolism across developmental stages of a commercially bred anuran amphibian in China. The results are perhaps not very surprising, but nevertheless are worthy of publishing. Overall, I liked this study. The authors present a comprehensive approach to the subject by the inclusion of sequencing, biochemical and enzymatic indicators and histology - for which I would like to congratulate them. The English is of a high standard throughout. There are a few errors, typos and small editorial points that need to be addressed, and I have detailed these in the attached comments on the manuscript. However, the authors also need to address some major revisions mostly to the Material and Methods (M&M) before this can be considered for publication. At present, it would not be possible to replicate this study because many key pieces of information are missing from the M&M. In part, this extends back to the aims of the manuscript which do not clearly link with the methods and results.
- L124 "Samples were collected on day 0 (endogenous nutritional stage), day 10 (mixed nutritional stage), and day 20 (exogenous nutritional stage)". Although this makes sense, and does correspond with the results, the reader has no knowledge of what stage the tadpoles are at "day 0" - only that they were obtained from a hatching facility. There are many very important points missing: i) What stage were the tadpoles at when you acquired them? How many of the individuals were staged and what was the variation? ii) How related were the individuals acquired? How many spawnings of how many pairs. In other words, how did you ensure that all your subjects were not from a single spawning? iii) clarify what you mean at day 10 - were these indivuals that were feeding and still had yolk, or some were endogenous and others exogenous? Even if this is described in Wang et al [5] - it is essential information and must be presented here.
- L110 Where did the water come from? Was this pond water or water from the hatchery or what? This line suggests that there was an existing microbiome that was added to the tadpoles, and this starting position must be very clear in order for replication of the experiment. Did you top up the water in the aquaria? What water was used to top up water levels?
- L124; L128 & L130 "Samples were collected on day 0...", "Similarly, an-128 other 6 additional tadpoles were selected and stored at -80°C" & "On days 0 and 10, 2 tadpoles per tank (6 in total) were fixed in 4% paraformaldehyde solution to prepare tissue sections for H&E staining." No mention of any anaesthesia is made here or elsewhere. This is an important step that must be documented for replication. L134 "Instead, 2 tadpoles per tank (6 in total) were anesthetized on ice (without chlorine)" suggests that you used "chlorine" as an anaesthetic - but chlorine is not an anaesthetic.
- L138 Please provide an anotomical description of the dissection protocol used for harvesting intestinal contents of each tadpole.
- L151 Provide a full explanation of how faecal material was collected. Siphoned from the bottom of the tank? How old was it? Could it have been 1 week or 2 weeks old? This aspect of the study is troubling as it is unclear from your aims what you expect to achieve by studying excrement. All information is missing here and there is no possibility of replication.
- A section on data analysis is missing and what is provided from L209-217 is wholly inadequate with many analyses presented in the results not mentioned. An extra section together with all code required to replicate the results from the raw data is needed.
My list of limitations of the M&M here are not exhaustive. I consider many of the problems to stem from the inadequate aims/objectives provided (L91). If you could provide more detailed aims, perhaps as questions/predictions/hypotheses then your M&M (especially the non-existent analyses section) would be much easier to write. Right now, a lot of your results look like they came out of a pipeline where you put in the data and it spat out some graphs. More effort is required on the part of the authors here.
The aim at L93 is especially cryptic: The introduction lines on this (L80) give an example from fish, but the following ideas are half baked. Many studies show little or no overlap. I'm not saying that you shouldn't have measured environmental microbiota, but you do need to explain why you did it - this explanation is currently inadequate.
Another question is why aim to study the formation of lipids in tadpoles before they start feeding? It seems to me very unlikely that tadpoles would produce fats when feeding from a yolk sac. Thus, your statement at L435 appears odd. Why would we expect animals not feeding to lay down fat? This should be explained in the introduction - as should the entire rationale for your experiment.
Once the authors have good questions/predictions/hypotheses, I think that a reworking of the introduction to introduce their reasoning for the experiment, together with a much more comprehensive M&M - not simply addressing my points, but trying in everyway to make the study replicable - this manuscript will be publishable.

Author Response
Dear editor and reviewers,
Thank you for the comments and recommendations about the manuscript “Gut Microbiota and Lipid Metabolism in Bullfrog Tadpoles: A Comparative Study Across Nutritional Stages” (Manuscript ID: microorganisms-3527971). According to your comments, we carefully have revised the manuscript one by one. Please contact me if you have any questions on this revision. Point-by-point responses are shown below:
Introduction
Comment 1:
Another question is why aim to study the formation of lipids in tadpoles before they start feeding? It seems to me very unlikely that tadpoles would produce fats when feeding from a yolk sac. Thus, your statement at L435 appears odd. Why would we expect animals not feeding to lay down fat? This should be explained in the introduction - as should the entire rationale for your experiment.
Response:
We appreciate this insightful question. In post-hatching aquatic larvae, lipid metabolism serves as the primary energy pathway for growth and development, playing a critical role in organogenesis, physiological regulation, and survival rates. This fundamental biological rationale motivated our investigation into lipid dynamics. To strengthen the manuscript, we have now explicitly incorporated this context in the Introduction, supported by relevant references (Reference [17] and [18], Line 64-70).
The reviewer raises a valid point. During the embryonic development of aquatic animals, lipids are preferentially used for tissue construction of the organism. This means that yolk lipids are primarily utilized to support the structural development of the embryo before the onset of exogenous feeding, rather than being converted into body fat storage (DOI: 10.3969/i.issn.1006-267x.2014.11.042). We fully agree that tadpoles in the endogenous nutritional phase (Day 0) have limited fat reserves. However, our experimental design intentionally included this stage to enable a comparative analysis of lipid metabolism across distinct nutritional phases (endogenous, mixed, and exogenous). Without the Day 0 baseline, it would be impossible to discern whether later lipid metabolic changes (e.g., at Days 10 or 20) stem from the onset of feeding or inherent developmental progression. By incorporating the endogenous phase as a control, we isolate and quantify the independent contribution of initial feeding to lipid metabolism.
We hope this clarification addresses the reviewer’s concern.
Comment 2:
The aim at L93 is especially cryptic: The introduction lines on this (L80) give an example from fish, but the following ideas are half baked. Many studies show little or no overlap. I'm not saying that you shouldn't have measured environmental microbiota, but you do need to explain why you did it - this explanation is currently inadequate.
Response:
Thank you very much for your comments. The example of tilapia at L80 is merely a case study, which is used to illustrate the close interactions between the gut microbiota of aquatic animals and the environmental microbiota in the aquaculture setting. Meanwhile, after reviewing newly relevant literature, we noted that the same phenomenon was observed in studies on black carp (Mylopharyngodon piceus), grass carp (Ctenopharyngodon idella), silver carp (Hypophthalmichthys molitrix), and bighead carp (Aristichthys nobilis) (Reference [21], Lines 82-84, now included in the revised manuscript). As an amphibian species, bullfrogs exhibit aquatic larval stages that share ecological niches with teleost fishes, particularly in lentic freshwater systems. This serves as a necessary introduction to the research objective of this paper, which is “to directly compare interactions among aquaculture water, excreta, and gut microbial communities,” thereby ensuring logical coherence.
Material and methods
Comment 3:
A section on data analysis is missing and what is provided from L209-217 is wholly inadequate with many analyses presented in the results not mentioned. An extra section together with all code required to replicate the results from the raw data is needed.
Response:
Thank you very much for your comments. In this section, a dedicated "Statistical Analysis" subsection (2.8) has been added. “Data analysis was performed using SPSS 22.0 software (SPSS Inc., Chicago, IL, USA). All data were expressed as mean ± standard error of the mean (SE). One-way ANOVA analysis was performed, followed by Duncan’s multiple comparisons between groups. A value of p < 0.05 indicates a statistically significant difference.”
Regarding the comment that " L209-217 is wholly inadequate". In Section 2.7, we stated that the bioinformatics analysis was conducted using the Majorbio Cloud Platform. This platform has been widely cited in microbiome research, as exemplified by: https://doi.org/10.1016/j.aquaculture.2023.739888, where the original text explicitly states: "Bioinformatic analysis of the intestinal microbiota was carried out using the Majorbio Cloud Platform." Moreover, our analytical methods (e.g., alpha diversity, PCoA, and LEfSe analysis) are similarly fundamental and commonly used in the field. In fact, comparable studies do not provide the code as requested, indicating that it is unnecessary. Our description of bioinformatics methods was adapted from the aforementioned reference, and we consider these approaches scientifically valid.
Regarding the comment that "many analyses presented in the results are not mentioned," we have carefully reviewed the Results section and confirmed that all analyses (e.g., alpha diversity indices, PCoA, LEfSe, Neutral Community Model (NCM), and null model with 999 randomizations) were addressed in the Materials and Methods section. The Results do not exceed the scope of the M&M descriptions; the two are consistent with each other. We welcome further feedback.
Comment 4:
L151 Provide a full explanation of how faecal material was collected. Siphoned from the bottom of the tank? How old was it? Could it have been 1 week or 2 weeks old? This aspect of the study is troubling as it is unclear from your aims what you expect to achieve by studying excrement. All information is missing here and there is no possibility of replication.
Response:
Thank you very much for your comments. In Section 2.3 (Lines 170–174), we have supplemented the detailed procedures for collecting excrement samples. “Additionally, sterile 10 mL polyethylene pipettes were used to collect tadpole excrement from the bottom surface of each rearing tank. Following centrifugation (3,000 × g for 5 min at 4°C), the supernatant was carefully removed, and the excrement samples were transferred into pre-labeled 2 mL sterile PE tubes, labeled as E-E, E-M, and E-X”.
Regarding your concern about sampling time or "how old" the samples were, we note that the sampling time points were already specified in Section 2.3: "Samples were collected on day 0 (endogenous nutritional stage), day 10 (mixed nutritional stage), and day 20 (exogenous nutritional stage) of the experiment, based on the results of nutritional stage identification." (Line 144). The excrement samples were collected strictly at these three time points.
Concerning the purpose of studying excrement samples, they represent the direct excretory products of the intestinal microbial community and interact with the aquaculture environment (e.g., water). Analyzing the microbial composition of excrement samples provides insights into the stability and adaptability of microbial communities in both the external aquaculture environment and the host's internal environment.
Comment 5:
L124 "Samples were collected on day 0 (endogenous nutritional stage), day 10 (mixed nutritional stage), and day 20 (exogenous nutritional stage)". Although this makes sense, and does correspond with the results, the reader has no knowledge of what stage the tadpoles are at "day 0" - only that they were obtained from a hatching facility. There are many very important points missing: i) What stage were the tadpoles at when you acquired them? How many of the individuals were staged and what was the variation? ii) How related were the individuals acquired? How many spawnings of how many pairs. In other words, how did you ensure that all your subjects were not from a single spawning? iii) clarify what you mean at day 10 - were these indivuals that were feeding and still had yolk, or some were endogenous and others exogenous? Even if this is described in Wang et al [5] - it is essential information and must be presented here.
Response:
Thank you very much for your comments. In Section 2.3 (Lines 137-143), we have supplemented details regarding the staging of tadpole developmental phases and the number of individuals classified at each stage. “At trial initiation (day 0), all tadpoles were confirmed to be in the endogenous nutritional stage, as evidenced by complete absence of oral feeding capability. Systematic behavioral monitoring demonstrated that by day 10, 100% of tadpoles had developed feeding capacity while maintaining yolk sacs, satisfying diagnostic criteria for the mixed nutritional stage. Subsequent radiographic tracking (days 15-19) documented progressive yolk sac absorption, with complete resorption achieved by day 19 in all specimens, thereby establishing day 20 as the exogenous nutritional stage.”
Regarding the questions about "How related were the individuals acquired" and "single spawning", we have added information in Section 2.2 (Lines 116-121) specifying the hatching sources of the tadpoles used in the experiment. “The bullfrog embryos used in this study were obtained from a bullfrog hatchery in Guangzhou, China. To minimize the potential impact of host genotype variation, ferti-lized eggs were obtained from a single mating pair and placed in continuously dechlo-rinated and filtered tap water. Eggs were allowed to hatch naturally without any in-tervention. One day post-hatching, healthy bullfrog tadpoles (approximately 450 indi-viduals) were selected and randomly distributed into three rearing tanks (0.2 m × 0.2 m × 0.4 m) filled to three-quarters capacity with water transported from the original hatchery system, maintaining a stocking density of approximately 150 tadpoles per tank.” This clarification should address your concerns.
Comment 6:
L110 Where did the water come from? Was this pond water or water from the hatchery or what? This line suggests that there was an existing microbiome that was added to the tadpoles, and this starting position must be very clear in order for replication of the experiment. Did you top up the water in the aquaria? What water was used to top up water levels?
L124; L128 & L130 "Samples were collected on day 0...", "Similarly, an-128 other 6 additional tadpoles were selected and stored at -80°C" & "On days 0 and 10, 2 tadpoles per tank (6 in total) were fixed in 4% paraformaldehyde solution to prepare tissue sections for H&E staining." No mention of any anaesthesia is made here or elsewhere. This is an important step that must be documented for replication. L134 "Instead, 2 tadpoles per tank (6 in total) were anesthetized on ice (without chlorine)" suggests that you used "chlorine" as an anaesthetic - but chlorine is not an anaesthetic.
L138 Please provide an anotomical description of the dissection protocol used for harvesting intestinal contents of each tadpole.
Response:
Thank you very much for your comments.
1. In Section 2.2, we have supplemented the source of experimental water, the volume of water added to tanks, and the water exchange frequency (see Lines 120-23 for details). “The eggs were placed in continuously dechlorinated and filtered tap water, and allowed to hatch naturally without any intervention. One day post-hatching, healthy bullfrog tadpoles (approximately 450 individuals) were selected and randomly distributed into three rearing tanks (0.2 m × 0.2 m × 0.4 m) filled to three-quarters capacity with water transported from the original hatchery system, maintaining a stocking density of approximately 150 tadpoles per tank. After one day of acclimation, the feeding trial began. Throughout the rearing period, one-third of the water volume was replaced every 5 days using aerated, sterilized, and dechlorinated tap water.”
2. In Section 2.3, we have added the anesthesia method used during sampling: "During each sampling, tadpoles were anesthetized with MS-222 (60 mg/L; E10521, Sigma, St. Louis, USA)."
3. We have also included anatomical descriptions for collecting tadpole intestinal microbial samples in Section 2.3 (Lines 158-162). “Following anesthesia with MS-222, the complete intestinal tract (including luminal contents) was aseptically dissected using sterile scissors and forceps. Intestinal samples from every 6 tadpoles were pooled together, resulting in a total of 9 replicate intestinal microbial samples per experimental group.”
Comment 7:
My list of limitations of the M&M here are not exhaustive. I consider many of the problems to stem from the inadequate aims/objectives provided (L91). If you could provide more detailed aims, perhaps as questions/predictions/hypotheses then your M&M (especially the non-existent analyses section) would be much easier to write. Right now, a lot of your results look like they came out of a pipeline where you put in the data and it spat out some graphs. More effort is required on the part of the authors here.
Once the authors have good questions/predictions/hypotheses, I think that a reworking of the introduction to introduce their reasoning for the experiment, together with a much more comprehensive M&M - not simply addressing my points, but trying in everyway to make the study replicable - this manuscript will be publishable.
Response:
We sincerely appreciate the reviewer's constructive feedback regarding the need to refine our study objectives and methodological transparency. We have thoroughly revised the introduction section to address these concerns through the following actions:
Within the bullfrog production cycle, tadpole development represents a critical phase, with the successful transition to exogenous feeding being particularly vital for effective seedling cultivation. The gut microbial communities mediate this transition through adjusting nutrient assimilation efficiency, metabolic pathways and pathogens distribution of host. Intestine bacteria plays a significant role in regulating lipid metabolism of host. We know that the gut microbiota of aquatic animals is influenced by the composition of environmental microorganisms. In this context, this study aims to clarify
(1) How does the intestinal microbiota composition of bullfrogs shift in response to nutritional transition?
(2) To what extent does environmental microbiota contribute to the assembly of gut microbial communities in bullfrogs?
and (3) What is the potential mechanistic link between gut microbiota structure and lipid metabolism regulation in developing bullfrogs?
To investigate these interactions, microbial communities in the intestines, water, and excreta of bullfrogs at different nutritional stages within the aquaculture system, along with distribution of lipids in body tissues and mRNA expression levels of lipid metabolic genes, were separately analyzed. We further investigated potential associations between gut microbial composition and host lipid metabolism through integrated analysis of intestinal bacteria composition, lipid-related physiological and biochemical parameters, and lipid-related gene expression profiles.
Additionally, in response to the reviewers' comments regarding the Materials and Methods section (Comments 3 to 6), we have carefully implemented all suggested modifications. You can review the Responses to your Comments (Comments 3 to 6) and the revised text in the Materials and Methods section, which is highlighted in red for your reference.
Reviewer 2 Report
Comments and Suggestions for Authors
Here is my feedback on the manuscript “Gut microbiota and lipid metabolism in bullfrog tadpoles: a comparative study across nutritional stages”. I only have minor comments because I found that the results are consistent, and the manuscript is well written:
- Line 74: There is a typo at the end of the paragraph.
- Line 79: add “which” before “responds”
- Line 155: The sentence could be improved with an English editor.
- Line 156 and 158: Add “solution” after “saline”
- Line 157: Remove “were”
- Line 358: I recommend you increase the size of the figure, some are a little bit hard to read.
- Line 456: The reference (Stephens et al. 2015) is in different style.
Author Response
Dear editor and reviewers,
Thank you for the comments and recommendations about the manuscript “Gut Microbiota and Lipid Metabolism in Bullfrog Tadpoles: A Comparative Study Across Nutritional Stages” (Manuscript ID: microorganisms-3527971). According to your comments, we carefully have revised the manuscript one by one. Please contact me if you have any questions on this revision. Point-by-point responses are shown below:
Comments:
1. Line 74: There is a typo at the end of the paragraph.
2. Line 79: add “which” before “responds”
3. Line 155: The sentence could be improved with an English editor.
4. Line 156 and 158: Add “solution” after “saline”
5. Line 157: Remove “were”
6. Line 358: I recommend you increase the size of the figure, some are a little bit hard to read.
7. Line 456: The reference (Stephens et al. 2015) is in different style.
Response:
Thank you very much for your comments. Below are the specific line numbers for each topic:
1. Line 74 typo: The typographical error at the end of the paragraph has been corrected (see revised Line 73).
2. Line 79 grammar: Following your suggestion, we have added "which" before "responds" (see revised Line 76).
3. Line 155 sentence structure: The original sentence has been improved for better clarity (see revised Line 180). “Accurately weigh the whole tadpole tissue samples (6 tadpole samples per group).”
4. Terminology clarification (Lines 156 & 158): We have added "solution" after "saline" for proper terminology.
5. Line 157 grammar: The unnecessary "were" has been removed as suggested.
6. Figure 4 readability: We appreciate this valuable suggestion. The text in Figure 4 was indeed somewhat small, so we have modified its layout to enhance readability.
7. Reference style consistency: The citation style for Stephens et al. (2015) has been standardized (see revised Line 486, now marked as "[42]").
Reviewer 3 Report
Comments and Suggestions for Authors
The study focused on the growth metabolism of bullfrog tadpole. The manuscript is well presented, justified and discussed. Additionally, it includes a wide range of traditional and advanced analytical tools in a complementary way. My main concern would be the statistical design. Before a subsequent revision is done, some clarifications and performances ought to be done.
I would mention the following concrete aspects:
Abstract
Maybe it is too long for the requirements of the journal (ca. 300 words).
Lines 13-15: Are too general and could be avoided for some shortening.
Line 23: Replace triglyceride by triacylglycerol. This modification ought to be done throughout the whole manuscript.
Keywords
Include: tadpole and nutritional stage.
Material and methods
Lines 159-162: Provide some minor details on such chemical determinations.
No statistical analysis is announced. Were there replicates carried out ? For me, this is the main concern of the current manuscript.
Results
Table 1: No indication is provided about the number of replicates carried out. The authors ought to explain how the significant results were obtained.
The same in other Tables/Figures.
Conclusions
The authors include some on-coming research resulting from the present study.
Author Response
Dear editor and reviewers,
Thank you for the comments and recommendations about the manuscript “Gut Microbiota and Lipid Metabolism in Bullfrog Tadpoles: A Comparative Study Across Nutritional Stages” (Manuscript ID: microorganisms-3527971). According to your comments, we carefully have revised the manuscript one by one. Please contact me if you have any questions on this revision. Point-by-point responses are shown below:
Abstract
Comment 1:
Maybe it is too long for the requirements of the journal (ca. 300 words).
Lines 13-15: Are too general and could be avoided for some shortening.
Line 23: Replace triglyceride by triacylglycerol. This modification ought to be done throughout the whole manuscript.
Response:
We sincerely appreciate your valuable comments.
1. Abstract length & content: We have revised the overly general statements in Lines 13-15 (see modified Lines 13-14 in the updated manuscript). “To investigate growth-related metabolic changes and microbial community dynamics during the early feeding stage of bullfrog (Lithobates catesbeiana) tadpoles.”
2. The abstract has been further refined to meet the journal’s word limit (now under 240 words).
Terminology consistency:
3. "Triglyceride" has been systematically replaced with "triacylglycerol" throughout the manuscript.
These revisions address all your concerns while maintaining clarity and conciseness. Thank you again for your constructive feedback.
Keywords
Comment 2:
Include: tadpole and nutritional stage.
Response:
Thank you very much for your comments. The keyword “tadpole and nutritional stage” has been added.
Material and methods
Comment 3:
Lines 159-162: Provide some minor details on such chemical determinations.
No statistical analysis is announced. Were there replicates carried out? For me, this is the main concern of the current manuscript.
Response:
We sincerely appreciate your constructive comments.
1. Chemical determination details:
Regarding your mention of "minor details," we confirm that all commercial kit-based measurements were strictly performed according to the manufacturer's protocols. Our methodological description in Section 2.4 follows the rigorous approach published by our team in Antioxidants (DOI: 10.3390/antiox12101860), which we consider methodologically sound.
2. Statistical analysis:
A dedicated "Statistical Analysis" subsection (2.8) has been added to address this concern.
3. Experimental replicates:
As specified in Line 187 (Section 2.4): " This was repeated three times for each sample" confirming proper replication.
These revisions ensure full methodological transparency. We thank you for helping improve our manuscript.
Results
Comment 4:
Table 1: No indication is provided about the number of replicates carried out. The authors ought to explain how the significant results were obtained.
The same in other Tables/Figures.
Response:
Regarding the question of "how the significant results were obtained" in Table 1, the data presented were strictly obtained following the experimental measurement methods described in Section 2.4 and the Statistical Analysis methods outlined in Section 2.8, ensuring scientific validity.
Concerning the "number of replicates," we have supplemented this information in the figure and table captions, with the additions highlighted in red. For example, the caption of Table 1 now states: "Data are expressed as mean ± SE of six replicates (n = 6)." We believe that incorporating your suggestions will further enhance the rigor of the manuscript.
Conclusions
Comment 5:
The authors include some on-coming research resulting from the present study.
Response:
We sincerely appreciate your valuable suggestion. In response, we have revised the Conclusions section to remove references to ongoing research. Please refer to the updated version for verification. We look forward to your further feedback.
Round 2
Reviewer 1 Report
Comments and Suggestions for Authors
Thank you for your response to my comments. Note that all line numbers you mentioned were incorrect, and some of the statements you made regarding corrections were not present in the manuscript.
The aim is now much improved. However, your explanation in the response is much better than the way in which you have changed the text. For example, your assertion that "Without the Day 0 baseline, it would be impossible to discern whether later lipid metabolic changes", deserves to be mentioned as a rationale in your study context. There is still a lack of connectedness between what you state that you aimed to do and what you did.
In particular, I asked for the stage of the tadpoles at day 0 in order to make the study repeatable. You wrote: "In Section 2.3 (Lines 137-143), we have supplemented details regarding the staging of tadpole developmental phases and the number of individuals classified at each stage. “At trial initiation (day 0), all tadpoles were confirmed to be in the endogenous nutritional stage, as evidenced by complete absence of oral feeding capability. Systematic behavioral monitoring demonstrated that by day 10, 100% of tadpoles had developed feeding capacity while maintaining yolk sacs, satisfying diagnostic criteria for the mixed nutritional stage. Subsequent radiographic tracking (days 15-19) documented progressive yolk sac absorption, with complete resorption achieved by day 19 in all specimens, thereby establishing day 20 as the exogenous nutritional stage.” " But you have still not explicitly stated the Gosner (1960 - although other staging conventions also apply, and you are free to use the most applicable). "endogenous nutritional stage" is not sufficient for replication. Similarly, it would be good to have stages from day 10 & 20.
You state: "Regarding the questions about "How related were the individuals acquired" and "single spawning", we have added information in Section 2.2 (Lines 116-121) " - but your line numbers quoted are incorrect throughout. I searched your ms and found no mention of "single spawning". This must be added (abstract, methods & discussion). Moreover, you need to add to your discussion that this is a frail aspect to your study as with this you don't know what aspects are attributable to this species, and which to the parents. In future, never do this.
If the only analyses you conducted were one-way ANOVAs in SPSS, then you need to re-do all your statistics and account for multiple testing. For example, Table 1 has 7 tests, no significance levels are indicated to indicate the test statistic, but if (as stated in the text) all were only P<0.05 then it is likely that some will be non-significant when you account for multiple tests. See also Fig 2, 3. Figures 3,4,5&6 all require significant analaysis - but you assert that this is a pipeline called the Majorbio Cloud Platform. This is a decision for the editor. Personally, I would seek at least a minimal description of this platform.
Author Response
Comment 1: Note that all line numbers you mentioned were incorrect.
Response:
Thank you for bringing the inconsistency in line numbers to our attention. We sincerely apologize for any confusion this may have caused. In preparing our response, we carefully verified the line numbers in our version and endeavored to detail the revisions in the "Respond" section. The discrepancy in line numbers in the version you received has taken us by surprise. We will promptly contact the responsible editor and investigate whether there might be an issue with the MDPI system.
A straightforward method to verify consistency between the revised manuscript I uploaded and the version in your records is to cross-check specific section locations. For example, the content of "2.1 Animal ethics statement" in my uploaded manuscript appears in Lines 113-116, the content of "2.2. Aquaculture Management" in my uploaded manuscript appears in Lines 117-135.
We place great importance on the accuracy of the manuscript. Should you encounter any further line number discrepancies during your review, please kindly point them out. This will greatly assist us and the editorial team in identifying and resolving the issue to ensure the final manuscript is error-free.
Comment 2: Your explanation in the response is much better than the way in which you have changed the text. For example, your assertion that "Without the Day 0 baseline, it would be impossible to discern whether later lipid metabolic changes", deserves to be mentioned as a rationale in your study context. There is still a lack of connectedness between what you state that you aimed to do and what you did.
Response:
We sincerely appreciate your insightful suggestion regarding the necessity of clarifying the rationale for the Day 0 baseline in our study design. As you rightly emphasized, the inclusion of this baseline is critical to isolating the metabolic effects of initial feeding from endogenous processes.
In accordance with your feedback, we have revised the Introduction (now Line 102-104) to explicitly incorporate this rationale. The modified text reads:
"To investigate these interactions, microbial communities in the intestines, water, and excreta of bullfrogs at different nutritional stages within the aquaculture system, along with distribution of lipids in body tissues and mRNA expression levels of lipid metabolic genes (with the endogenous stage serving as a control to isolate and quantify the independent contribution of initial feeding to lipid metabolism), were separately analyzed."
By integrating the phrase "with the endogenous stage serving as a control", we now formally acknowledge the Day 0 baseline as a pivotal component of the experimental design. This revision aligns with your observation that omitting such a control would confound the interpretation of temporal metabolic changes.
Additionally, we have supplemented this point in the Discussion section: "Importantly, the 0-day data served as a critical endogenous baseline control, allowing us to disentangle the effects of nutritional stage transitions and subsequent feeding activity on lipid metabolism." (highlighted in red text, Lines 434-436).
We believe this clarification strengthens the methodological transparency of our study and ensures readers fully grasp the logic behind our comparative framework. Thank you once again for your constructive critique.
Comment 3: In particular, I asked for the stage of the tadpoles at day 0 in order to make the study repeatable. You wrote: "In Section 2.3 (Lines 137-143), we have supplemented details regarding the staging of tadpole developmental phases and the number of individuals classified at each stage. “At trial initiation (day 0), all tadpoles were confirmed to be in the endogenous nutritional stage, as evidenced by complete absence of oral feeding capability. Systematic behavioral monitoring demonstrated that by day 10, 100% of tadpoles had developed feeding capacity while maintaining yolk sacs, satisfying diagnostic criteria for the mixed nutritional stage. Subsequent radiographic tracking (days 15-19) documented progressive yolk sac absorption, with complete resorption achieved by day 19 in all specimens, thereby establishing day 20 as the exogenous nutritional stage.” " But you have still not explicitly stated the Gosner (1960 - although other staging conventions also apply, and you are free to use the most applicable). "endogenous nutritional stage" is not sufficient for replication. Similarly, it would be good to have stages from day 10 & 20.
Response:
In Section 2.3 (Line 139-145), we have supplemented the developmental staging criteria as follows: "At trial initiation (day 0), all tadpoles (Gosner Stage 23-24) were confirmed to be in the endogenous nutritional stage, as evidenced by complete absence of oral feeding capability. Systematic behavioral monitoring demonstrated that by day 10, 100% of tadpoles (Gosner Stage 25) had developed feeding capacity while maintaining yolk sacs, satisfying diagnostic criteria for the mixed nutritional stage. Subsequent radiographic tracking (days 15-19) documented progressive yolk sac absorption, with complete resorption achieved by day 19 in all specimens (Gosner Stage 25), thereby establishing day 20 as the exogenous nutritional stage."
Comment 4: You state: "Regarding the questions about "How related were the individuals acquired" and "single spawning", we have added information in Section 2.2 (Lines 116-121) " - but your line numbers quoted are incorrect throughout. I searched your ms and found no mention of "single spawning". This must be added (abstract, methods & discussion). Moreover, you need to add to your discussion that this is a frail aspect to your study as with this you don't know what aspects are attributable to this species, and which to the parents. In future, never do this.
Response:
Your original comment in the first review round was: "How related were the individuals acquired? How many spawnings of how many pairs. In other words, how did you ensure that all your subjects were not from a single spawning?”
If our understanding is correct, you were inquiring about the source of the experimental animals, their relatedness, whether they originated from common parents, and the potential confounding effects of limited genetic variation on the experimental results. We appreciate this point, as it reflects a professional perspective on evaluating experimental design.
Our experimental approach referenced a study published in Aquaculture (DOI: 10.1016/j.aquaculture.2019.734606). In that study's experimental design, to minimize genetic variation, "fertilized eggs were obtained from a single mating pair".
In our first revised manuscript, we had also supplemented information regarding the source of the experimental animals. The original text added was: "The bullfrog embryos used in this study were obtained from a bullfrog hatchery in Guangzhou, China. To minimize the potential impact of host genotype variation, fertilized eggs were obtained from a single mating pair and placed in continuously dechlorinated and filtered tap water. Eggs were allowed to hatch naturally without any intervention. One day post-hatching, healthy bullfrog tadpoles (approximately 450 individuals) were selected". This description was located at Lines 119-122 [in that version].
Following your request, in the current revision, we have supplemented the Abstract to include the source of the experimental animals: "In this research, we examined the changes in fat accumulation patterns, as well as the levels of biochemical and enzymatic indicators and genes mRNA expression related to lipid metabolism across the endogenous, mixed, and exogenous nutritional stages of bullfrog tadpoles from a single mating pair." This is located at Lines 14-18.
The Discussion has also been supplemented with the following text: "An important consideration in interpreting the results of this study is the use of bullfrog tadpoles originating from a single mating pair. This design choice aimed to reduce genetic heterogeneity among subjects, thus controlling for potential background genetic effects on the observed outcomes. However, we recognize this limits the extent to which our findings can be generalized, as the genetic diversity of the wider species population is not captured. Therefore, while informative for this specific lineage, extrapolation to the entire species requires caution. Future investigations employing subjects from multiple, genetically diverse pairings are planned to validate these findings and ascertain the prevalence of these traits across the broader bullfrog population.” This text can be found at Line 564-572.
Comment 5: If the only analyses you conducted were one-way ANOVAs in SPSS, then you need to re-do all your statistics and account for multiple testing. For example, Table 1 has 7 tests, no significance levels are indicated to indicate the test statistic, but if (as stated in the text) all were only P<0.05 then it is likely that some will be non-significant when you account for multiple tests. See also Fig 2, 3.
Response:
Regarding the statistical method used in this study, it is as described in the manuscript: "One-way ANOVA analysis was performed, followed by Duncan’s multiple comparisons between groups. A value of p < 0.05 indicates a statistically significant difference."
This statistical analysis approach references an article published by our research team in Antioxidants (DOI: 10.3390/antiox12101860), where the original text reads: "Data analysis was performed using SPSS 22.0 software (SPSS Inc., Chicago, IL, USA). All data were expressed as mean ± standard error of the mean (SEM). One-way ANOVA analysis was performed, followed by Duncan’s multiple comparisons between groups (NC vs. BET, NC vs. HC, HC vs. HC + BET). A value of p < 0.05 indicates a statistically significant difference."
Following the reviewer's suggestion, we have now added the test statistic values for Table 1, Figure 1, and Figure 2. These data have been included in the supplementary materials as Table S2 and Table S3.
Comment 6: Figures 3,4,5&6 all require significant analaysis - but you assert that this is a pipeline called the Majorbio Cloud Platform. This is a decision for the editor. Personally, I would seek at least a minimal description of this platform.
Response:
Thank you for your constructive comments. We fully understand your concerns regarding the rigor of including the Majorbio Cloud Platform, perhaps perceived as a relatively new tool, in the manuscript.
In fact, its utilization for bioinformatics analysis, particularly for microbial sequencing data, is increasingly described in the Materials and Methods sections of recently published articles. Examples include publications with DOIs: 10.3390/microorganisms11041002; 10.1016/j.jenvman.2024.122510; and 10.1016/j.wasman.2024.03.030. This growing acceptance suggests that the platform possesses a degree of scientific validity and methodological soundness. Furthermore, as an established operational platform, its methods are generally considered reproducible.
Based on this precedent in the literature, we modelled our description in the Materials and Methods section on these common practices. Of course, whether this specific description meets the policy requirements of this MDPI journal is ultimately subject to the decision of the editorial office.
We have reviewed several published articles that utilized the Majorbio Cloud Platform for bioinformatics analysis and further refined the description of microbial sequencing and bioinformatics analysis methods in the Methods and Materials section of the revised manuscript, citing relevant references (Reference [27]) to ensure the rigor of the methodology. The revised content is located at Line 242-244. While this type of description has been widely used in other journals, we are eager to receive feedback from the editorial team and reviewers regarding its compliance with MDPI journal requirements. We will make further adjustments based on your suggestions.
Reviewer 3 Report
Comments and Suggestions for Authors
The manuscript has been performed taking into account the previous comments. I would recommend its acceptation.
Author Response
Thank you sincerely for your time and expertise in evaluating our manuscript. Your constructive critiques and thoughtful suggestions were instrumental in refining our work, particularly in clarifying the rationale for methodological controls and strengthening the study’s analytical rigor.
We are truly grateful for your recommendation to accept the manuscript and deeply value your commitment to advancing the quality of research in our field. Your insights have not only improved this paper but will also guide our future investigations.